# Refining Orbits of Bright Binary Systems

Anatoly S. Miroshnichenko [1,*], Stephen Danford [1], Ivan L. Andronov [2], Alicia N. Aarnio [1], Duncan Lauer [1] and Holly Buroughs [1]

1 Department of Physics and Astronomy, University of North Carolina—Greensboro, Greensboro, NC 27402, USA

2 Department of Mathematica, Physics and Astronomy, Odessa National Maritime University, Mechnikova St. 34, 65029 Odessa, Ukraine

* Correspondence: a_mirosh@uncg.edu

**Abstract:** We obtained spectra of several bright binary systems ($\zeta$02 UMa, 2 Lac, and $\phi$ Aql), which were mostly observed with photographic plates and whose orbits were not determined very accurately. Each system was monitored for a few years with the 81-cm telescope equipped with an échelle spectrograph at the Three College Observatory in North Carolina. The spectra were taken in a wavelength range between 4000 and 7900 Å with a spectral resolution of $R \sim 12{,}000$. Radial velocity measurements were done using cross-correlation in selected spectral regions or by measuring positions of individual spectral lines. Refined orbits and stellar parameters are presented.

**Keywords:** spectroscopy; binary system; stars

## 1. Introduction

Binary stellar systems are ubiquitous in the Universe. Observations obtained at high spacial resolution, such as those reported by [1], as well as theoretical studies (e.g., ref. [2] and references therein) suggest that most stars with initial masses over $\sim$10 M$_\odot$ exist in gravitationally bound pairs or multiple systems. Binary systems that contain normal (not emission-line) stars can show spectral lines of either one (single-lined) or both (double-lined) components. Determining their orbital properties, such as periods, radial velocity (RV) amplitudes, and eccentricities, from spectroscopic observations allows us to measure masses of the components directly.

Dealing with single-lined binaries is more complicated, because in addition to the inclination angle of the system's rotational axis the orbital parameters need to be known for both components in order to determine their masses. If the inclination angle in a double-lined spectroscopic binary is unknown, one may determine the mass ratio of the pair from the inverse ratio of the RV amplitudes but it may not be possible to determine each individual mass. The latter is possible with the knowledge of the fundamental parameters (temperature and luminosity).

Although many bright binary systems were observed with spectrographs using photographic plates, orbital parameters of a number of them were not determined with a high accuracy because spectral line measurements were of a low accuracy and were in a limited region of the spectrum. We attempted to refine orbital solutions for several bright binaries using modern CCD spectroscopy. Here we report our results for three such binaries: $\zeta$02 UMa (HR 5055), 2 Lac (HR 8523), and $\phi$ Aql (HR 7610).

## 2. Previous Orbit Determinations for the Program Objects

### 2.1. $\zeta$02 UMa = HR 5055 = Mizar B

Mizar B is the fainter among the two binary systems visible with the naked eye as one star, Mizar. Both systems are spectroscopic binaries easily resolved with a small telescope. Mizar B ($V = 3.9$ mag) is a single-lined binary, whose RV variations were

first announced by Frost [3]. The first orbit determination attempt was made by Abt [4] based on 24 photographic spectra taken at the McDonald Observatory (Texas, USA) in June–July 1959. However, the orbital period coverage turned out to be too small, and a wrong period of 361 days was derived. A more thorough spectral study was conducted by Gutmann [5], who used 89 photographic spectra taken at the Dominion Astrophysical Observatory (DAO, Victoria, BC, Canada) between 1941 and 1963. These data covered several orbital cycles, but only 3 of them were covered in detail that was important due to a large orbital eccentricity. To the best of our knowledge, no new orbital studies have been reported since that time. As one can see in Figure 1, the data points are very scattered around the best fit and definitely required further refinement.

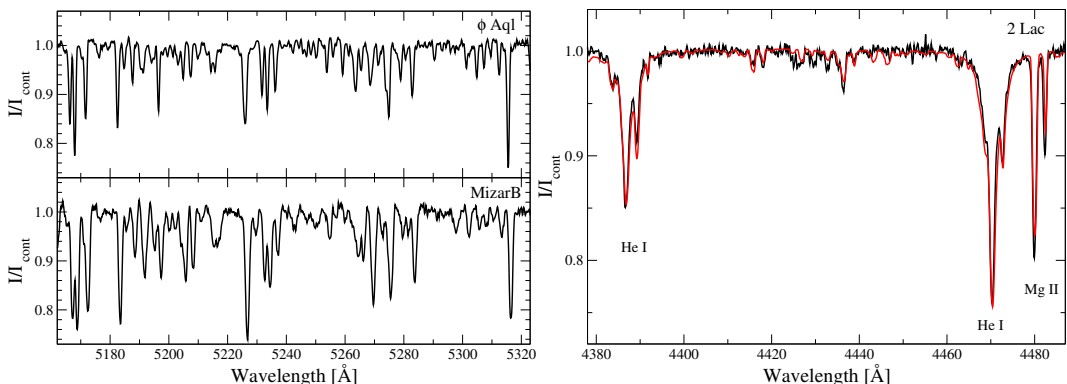

**Figure 1. Left panel:** Examples of the cross-correlated parts of the spectra of $\phi$ Aql and MizarB. Intensity is normalized to the local continuum. Wavelengths are shown in Å on a heliocentric scale. **Right panel:** Part of a spectrum of 2 Lac with a large RV separation (black line) and a model spectrum (red line) composed from two spectra calculated for $T_{eff}$ = 15,000 K, log g = 4.0, v sin $i$ = 49 km s$^{-1}$ shifted by $-85$ km s$^{-1}$ and for $T_{eff}$ = 16,000 K, log g = 4.0, v sin $i$ = 23 km s$^{-1}$ shifted by +80 km s$^{-1}$. The components' contributions to the total continuum intensity at 5550 Å are 80% and 20% for the primary and secondary, respectively. The modeling is described in Section 6.2.3.

### 2.2. $\phi$ Aquilae = HR 7610

$\phi$ Aquilae is a $V$ = 5.3 mag SB1 binary system. It was observed spectroscopically in 1920s–1930s by Harper [6,7] and listed in a catalog by Batten [8]. Lucy and Sweeney [9] recalculated the orbit using their method of testing for circular orbits.

### 2.3. 2 Lacertae = HR 8523

2 Lacertae is an SB2 binary system with components of nearly the same spectral type. Its observations were first reported by Baker [10] using 84 spectra taken at the Alegheny Observatory in 1908. Luyten et al. [11] obtained 41 spectra at the Yerkes Observatory in the 1930s. Finally, Hilditch [12] added 11 more spectra taken at DAO in the early 1970s and analysed the entire mentioned collection. Based on optical photometric data, Hilditch suggested that the secondary component is a B6 V star, while the primary component should be located near the end of core hydrogen burning to have about the same effective temperature ($T_{eff} \sim$ 15,000 K).

## 3. Observations and Data Reduction

Spectroscopic observations of MizarB, 2 Lac, and $\phi$ Aql were obtained mostly in 2018–2022, except for a few earlier observations, at the Three College Observatory (hereafter TCO), which is located in $\sim$12 km south of the city of Graham, Alamance county, North Carolina, USA. TCO has a 0.81 m telescope equipped with a fiber-fed échelle spectrograph from Shelyak Instruments[1]. The spectrograph with an ATIK-460EX detector (2749 × 2199 pixels, pixel size 4.54 μm × 4.54 μm) provides a spectral resolving power of $R \sim$ 12,000 in a spectral range from 3800 Å to 7900 Å without gaps between the spectral orders.

The Observatory was built at the end of the 1970s, while regular spectroscopic observations began at the end of 2011. The observational program includes a broad range of normal (with no emission lines) and emission-line stars, such as classical Be stars, objects with the B[e] phenomenon, Herbig Ae/Be stars, supergiants, etc. The described setup allows to take spectra of objects starting from the brightest stars (e.g., Sirius) to those of ∼10 visual magnitude. It takes a few seconds to reach a signal-to-noise ratio of 100–200 in the continuum for the brightest stars and 2–3 h to get it to ∼50 for the faintest. The highest signal-to-noise ratio (up to ∼300) is achieved in a spectral range between ∼4500 Å and ∼5500 Å.

Each spectrum typically consists of several individual exposures, which are summed up during the data reduction process that was done using the *echelle* package in IRAF. The latter includes bias subtraction, spectral order separation, and wavelength calibration using spectra of a ThAr lamp. Flat field images are not taken, because the detector has a pixel sensitivity difference of ≤1.5% and the flat field lamp does not cover the entire extracted spectral range. The spectra contained 24 orders in a region from 4200 Å to 7900 Å during the first few years of observations. In the Fall of 2018 the number of orders increased to 31 after the installation of an optical fiber with better UV transmission. There are no gaps in the wavelength coverage.

The number of comparison lines identified in the calibration spectra ranged from ∼800 to over 1000. Typical scatter of the comparison line positions from a polynomial solution is ∼0.03 Å which results in a RV accuracy of 300 m s$^{-1}$. This allows us to reliably detect RV variations of a few km s$^{-1}$. Recent examples of such results are studies of the Cepheid Polaris [13] and the binary system 3 Pup, in which the primary component shows the B[e] phenomenon [14].

The objects reported in this paper were observed as follows: individual spectra were taken with exposure times of 2–3 min for MizarB, 4–6 min for 2 Lac, and 8–10 min for $\phi$ Aql. Series of 3–10 exposures were taken in each observing session, summation of which resulted in typical signal-to-noise ratios of ∼200–300 at the 4000–5000 Å region and ∼100–200 at longer wavelengths. The wavelength scale was verified by observations of RV standard stars.

## 4. Results

The TCO RV data for Mizar B and $\phi$ Aql were derived by cross-correlation in a spectral region between ∼5100 and ∼5300 Å using the IRAF package *rvsao*. The spectra were cross-correlated against a template spectrum, which was averaged from several exposures of the RV standard star $\delta$03 Tau = HR 1389, heliocentric RV = 39.0 ± 0.1 km s$^{-1}$ [15,16]. Examples of the cross-correlated parts of the spectra of $\phi$ Aql and MizarB are shown in the left panel of Figure 1.

The RVs for the components of 2 Lac were measured using a Gaussian approximation of 10–15 absorption-line profiles between C II 4267 Å and He I 6678 Å lines. Some of these lines are marked in the right panel of Figure 1, which show the spectrum of the system taken on 4 November 2022 in the orbital phase 0.195, when the primary component's RV was shifted by −84.6 ± 4.3 km s$^{-1}$ and that of the secondary component was shift by +79.7 ± 6.8 km s$^{-1}$.

The orbital solutions were derived independently using two different codes written by I.A. and A.M. The resulting orbital parameters differed within the uncertainties, which were calculated by I.A. and briefly explained in the next Section. The derived parameters are listed in Tables 1–3 along with previously published solutions. The RV curves folded with the orbital periods are shown in Figures 2–4 along with the previously published data.

**Table 1.** Orbital elements derived for Mizar B (HR 5055).

| Element | [4] | [5] | TCO |
|---|---|---|---|
| P (days) | 361.24 | $175.55 \pm 0.06$ | $175.11 \pm 0.10$ |
| Epoch (HJD) | 2,417,623.68 | $2,437,294.4 \pm 1.6$ | $2,458,175.2 \pm 0.7$ |
| $e$ | 0.697 | $0.46 \pm 0.03$ | $0.56 \pm 0.09$ |
| $\omega$ (degrees) | 27.14 | $4.3 \pm 5.2$ | $353.0 \pm 4.6$ |
| $\gamma$ (km s$^{-1}$) | $-13.16$ | $-9.3 \pm 0.2$ | $-14.34 \pm 0.04$ |
| $K_1$ (km s$^{-1}$) | 6.73 | $6.3 \pm 0.2$ | $6.46 \pm 0.53$ |
| f(m), $M_\odot$ | 0.0042 | $0.0032 \pm 0.0005$ | $0.0028 \pm 0.0011$ |
| $N$ | 24 | 89 | 155 |

Parameters listed are as follows (line number): 1—orbital period, 2—periastron epoch, 3—eccentricity, 4—argument of the periastron, 5—systemic velocity, 6—semi-amplitude of the RV variation of the visible component, 7—mass function, and 8—number of spectra used in the orbit calculation. The parameters from Abt [4] contain no error determination. The parameters from Gutmann [5] were determined for the entire set of data, while in this paper results are also given for subsets.

**Table 2.** Orbital elements derived for $\phi$ Aquilae (HR 7610).

| Element | [6] | [9] | TCO |
|---|---|---|---|
| P (days) | $3.320 \pm 0.001$ | 3.320680 | $3.320669 \pm 0.000017$ |
| T0 (JD) | 2,423,324.045 | 2,423,210.628 | $2,459,445.0916 \pm 0.0011$ |
| $e$ | $0.055 \pm 0.022$ | 0 | 0 |
| $\omega$ (degrees) | 56.0 | − | − |
| $\gamma$ (km s$^{-1}$) | $-27.63 \pm 0.63$ | $-28$ | $-28.52 \pm 0.05$ |
| $K_1$ (km s$^{-1}$) | $38.25 \pm 0.72$ | 37.2 | $36.48 \pm 0.07$ |
| f(m), $M_\odot$ | $0.019 \pm 0.001$ | 0.018 | $0.0167 \pm 0.0001$ |
| N | 34 | 30 | 80 |

The parameters are the same as in Table 1 except for T0, which is the epoch when the visible component reaches the $\gamma$ RV moving away from the observer. The orbital elements given by Lucy and Sweeney [9] were recalculated using the data from Harper [7].

**Table 3.** Orbital elements derived for 2 Lacertae (HR 8523).

| Element | [12] | Recalculation | TCO |
|---|---|---|---|
| P (days) | $2.616430 \pm 0.000003$ | $2.611523 \pm 0.000004$ | $2.616535 \pm 0.000035$ |
| T0 (HJD) | $242,700.80 \pm 0.18$ | $2,422,741.402 \pm 0.011$ | $2,459,423.7130 \pm 0.0027$ |
| $e$ | $0.040 \pm 0.018$ | 0 | 0 |
| $\omega$ (deg) | $97.4 \pm 25.3$ | − | − |
| $\gamma$ (km s$^{-1}$) | $-8.9 \pm 1.1$ | $-10.01 \pm 1.38$ | $-13.23 \pm 0.34$ |
| $K_1$ (km s$^{-1}$) | $79.6 \pm 1.8$ | $72.78 \pm 1.91$ | $73.19 \pm 0.47$ |
| $K_2$ (km s$^{-1}$) | $100.0 \pm 1.8$ | − | $90.25 \pm 0.71$ |
| f(m$_1$), $M_\odot$ | $0.137 \pm 0.010$ | $0.104 \pm 0.007$ | $0.106 \pm 0.002$ |
| f(m$_2$), $M_\odot$ | $0.272 \pm 0.015$ | − | $0.199 \pm 0.004$ |
| N1/N2 | 82/61 | 135/63 | 42/29 |
| $m_1 \cdot \sin^3 i$, $M_\odot$ | $0.87 \pm 0.06$ | − | $0.653 \pm 0.021$ |
| $m_2 \cdot \sin^3 i$, $M_\odot$ | $0.69 \pm 0.05$ | − | $0.529 \pm 0.018$ |

Parameters in the upper 5 lines are the same as in Table 2. RV semi-amplitudes $K_1$ and $K_2$, mass functions f(m$_1$) and f(m$_2$) are listed for both components. N1 and N2 are the numbers of spectra used for the primary and secondary component orbital solution, respectively. N2 is typically lower than N1, because it is difficult to measure spectral line positions in the spectrum of the fainter secondary component near the phases close to the systemic RV, when the components' lines blend together. The lowest two lines show the calculated values of the components' masses multiplied by the cube of the system inclination angle. Third column shows the orbital parameters recalculated by us using all the data from [10–12].

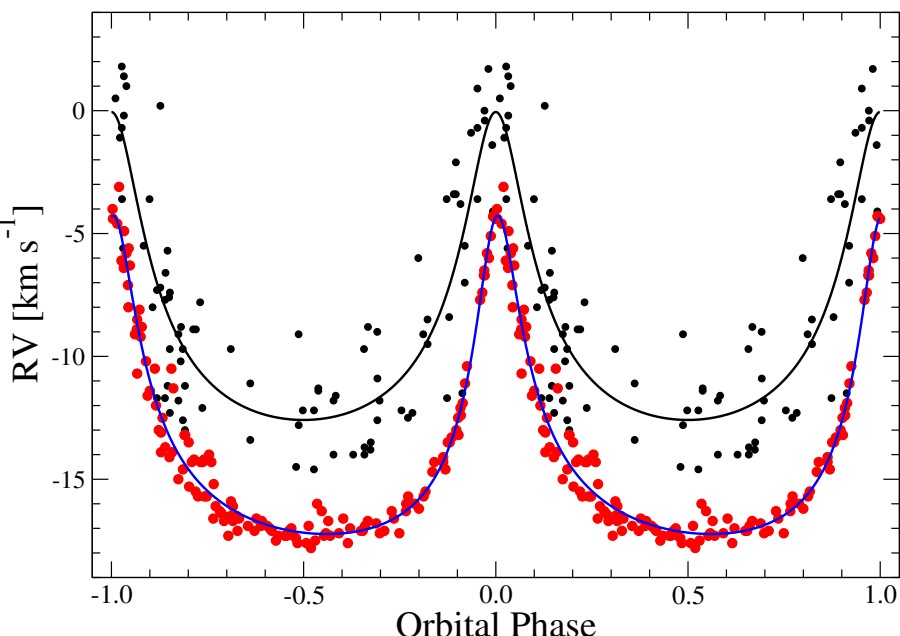

**Figure 2.** RV curves of Mizar B derived by Gutmann [5] and in this paper. The DAO data are shown by black dots, and the best fit to these data is shown by the black solid line. The TCO data are shown in red, and the best fit is shown by the blue solid line. The RVs are heliocentric.

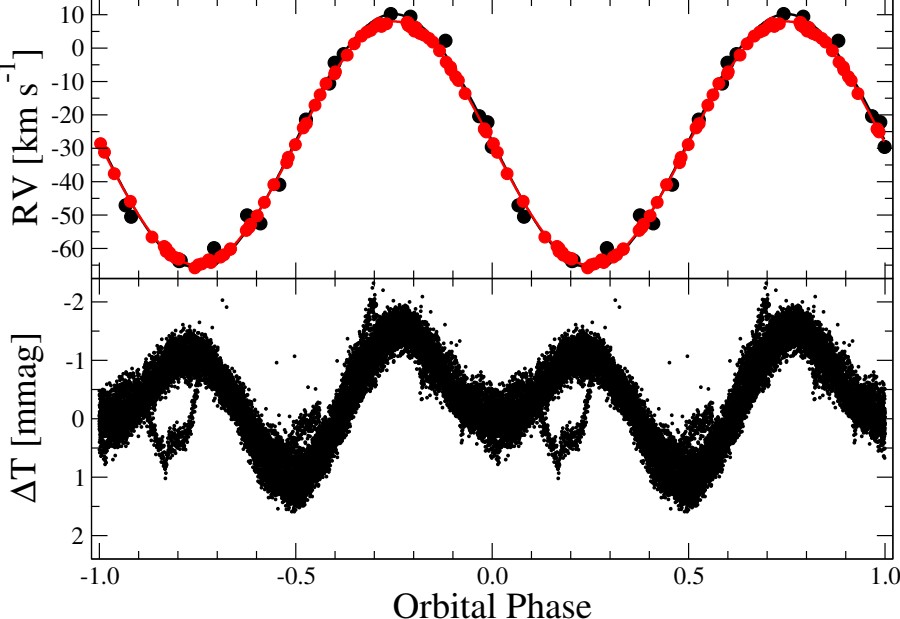

**Figure 3. Top panel:** Radial velocity curves of $\phi$ Aquilae plotted using data from Harper [6] and this paper. The DAO data are shown by black dots, and the best fit to these data is shown by the black solid line. The TCO data are shown in red, and the best fit is shown by the red solid line. The RVs are heliocentric. **Bottom panel:** TESS light curve folded with the orbital period. The relative flux scale is shown in millimagnitudes.

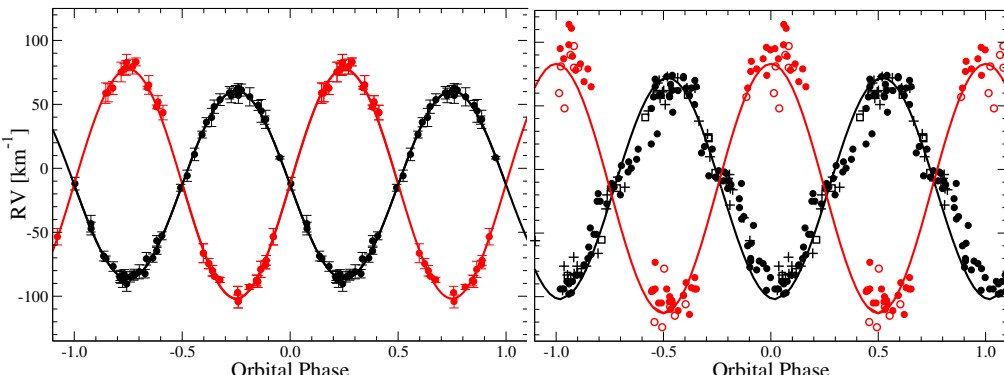

**Figure 4. Left panel** : RV curves of both components of 2 Lac plotted using TCO data. The data for the primary component are shown by black circles, and the best fit to these data is shown by the black solid line. The data for the secondary component are shown in red, and the best fit is shown by the red solid line. The RVs are heliocentric. **Right panel**: RV curve of both components of 2 Lac from [10–12] with data from individual papers shown by different symbols.

## 5. Derivation of the Orbital Parameter Errors

In addition to a more accurate method of the spectrum registration, methods of the data and error analysis have also changed in the last ~40–50 years (e.g., [17]). Moreover, since the error derivation is not described in most of the papers quoted here, it is important to explain our approach to the error calculation. This will also allow us to compare the published errors of the orbital parameters and those determined in this work.

The classical scheme of the least squares (LSQ) assumes the test function

$$\Phi(x) = \sum_{k=1}^{n} w_k \cdot (x_k - x_{C,k})^2, \tag{1}$$

where $x_k$ and $x_{C,k} = x_C(t_k)$ are the measured and "calculated" values at times $t_k$, respectively, and the approximation is

$$x_C(t) = \sum_{\alpha=1}^{m} C_\alpha \cdot f_\alpha(t), \tag{2}$$

$C_\alpha$ are coefficients (parameters), $f_\alpha(t)$ are basic functions, $w_k = \sigma_0^2/\sigma_k^2$ are "weights", $\sigma_k$ is the accuracy of $x_k$, and $\sigma_0$ is the "unit weight error". The latter parameter may be an arbitrary non-zero quantity, but for the error analysis it may be estimated as $\sigma_0^2 = \Phi_{min}/(n-m)$. The covariance matrix of the coefficients' errors is inversely proportional to the matrix of normal equations.

$$\langle C_{d,\alpha} C_{d,\beta} \rangle = \sigma_0^2 \cdot A_{\alpha\beta}^{-1} \tag{3}$$

and the coefficient uncertainties are

$$\sigma^2[C_\alpha] = \langle C_{d,\alpha} C_{d,\alpha} \rangle = \sigma_0^2 \cdot A_{\alpha\alpha}^{-1}, \tag{4}$$

see Anderson [18] and Press et al. [19] for details.

For a sinusoidal signal and its modifications, Andronov [20] developed a code, which improves the algorithm to non-linear least squares (NLLSQ, the trial frequency is an additional "non-linear" parameter) with the Newton–Raphson "differential corrections". In this case, the set of basic functions is extended to $M = m + 1$ as $f_\alpha(t) = \partial x_C(t)/\partial C_\alpha$ (see also [21] for a recent review). In particular, if one needs to replace one of the coefficients with $C_\alpha \pm \sigma[C_\alpha]$, then $\Phi = \Phi_{min} \cdot (1 + 1/(n-M))$. As a matter of fact, for large error

estimates of the "non-linear" parameter, one may look for a region with such an increased constant level.

An alternative method for the error estimates is called the "Bootstrap" method [22], where the data are randomly generated instead of real data (real data points are used with a random number). In other words, if some data points are absent (zero weight), they may occur more than once (their weights increase proportionally). Andrych et al. [23] applied this method to some astronomical data.

## 6. Discussion

### 6.1. Error Comparison

Comparison of the RV curves folded with the orbital periods shows a much lower scatter of the CCD data around the best fit for MizarB and 2 Lac compared to the photographic data (see Figures 1 and 3). The scatter of the $\phi$ Aql CCD data is marginally better, but the phase coverage is much more dense (see Figure 2). Nevertheless, the previously published errors of the orbital parameters are comparable and, in some cases, even smaller than those determined in the present paper.

Two factors that determine the parameter accuracy are the number of observations and the total time period the data cover. The latter definitely affected the orbital period determination of 2 Lac, where the photographic data sets covered nearly 6 decades, although with large gaps. Using the error calculation algorithms described in Section 5, we have computed the error estimates for the parameters of the RV approximation for the primary component of 2 Lac. The "best-fit" period from the "Bootstrap" method (10,000 random data points) is 2.616533 days, by 2 in the last digit smaller compared to the NLLSQ estimate. The average error is exactly the same (0.000036 days), but the (typical for the "Bootstrap" method) asymmetrical error estimates $\sigma_- = 0.000034$, $\sigma_+ = 0.000032$ are slightly smaller and comparable with those computed with the NLLSQ method. A similar behavior is seen for other orbital parameters. At the same time, the TCO errors of all the parameters but the period are noticeably smaller than those of the DAO observations.

The same applies to MizarB, where the DAO observations span $\sim$2 decades, while our data were taken within 4 years. Another feature of the MizarB DAO data set is systematically more positive RVs mentioned in [5] in comparison with the McDonald data published in [4]. Although Gutmann [5] presented some RV measurements of the standard star HD 91480 that are no different from the modern value of its RV $= -12.11 \pm 0.02\ \mathrm{km\,s^{-1}}$ [16], our data verified by using another standard star, HR 1389, are closer to the McDonald data.

Despite a lower scatter of our MizarB data, our orbital parameter errors turned out to be larger than those from [5]. One reason for that is a possible oversimplification of the error determination in [5], where the error determination method was not mentioned. Our error analysis of the photographic data resulted in a period error of 0.11 days, which is only slightly larger than that of our data but still about twice as large compared to the error published in [5]. The errors of the other parameters are either comparable or smaller for our data set. However, this may also point to a simplified error determination in [5].

### 6.2. Fundamental Parameters

The orbital parameters for $\phi$ Aql and 2 Lac derived from our data are definitely more accurate, except that for the orbital period of 2 Lac described above. In combination with the Gaia data for the distances toward the systems [24] and photometric data taken from various sources, our results allow us to derive fundamental properties of the systems' components. They are shown in Table 4 and in Figure 5.

**Table 4.** Fundamental parameters of the studied systems.

| Name | $V$–mag | D, pc | $T_{eff}$, K | log $L/L_\odot$ |
|---|---|---|---|---|
| MizarB | 3.91 | $24.81 \pm 0.17$ | $8279 \pm 58$ | $1.13 \pm 0.01$ |
| $\phi$ Aql | 5.28 | $67.67 \pm 0.35$ | $9484 \pm 132$ | $1.45 \pm 0.01$ |
| 2 Lac | 4.54 | $188.9 \pm 6.0$ | $13{,}996 \pm 97$ | $3.12 \pm 0.04$ |

$V$–band magnitudes are taken from [25], where they were measured with an accuracy of $\leq 0.01$ mag; distances are taken from [24]; effective temperatures are shown for the primary components and taken from [26], the luminosity is calculated using these data and bolometric corrections from [27]. The luminosity of 2 Lac refers to the combined light of both system components. The apparent visual magnitude of 2 Lac was corrected for an interstellar extinction of $A_V = 0.1$ mag for the luminosity calculation.

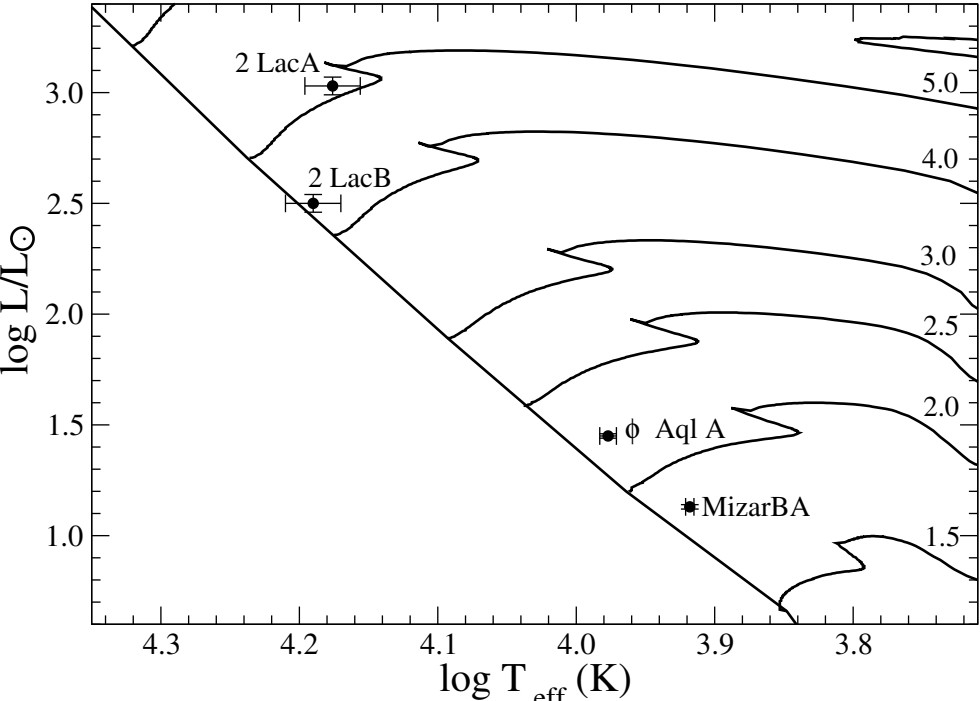

**Figure 5.** HR diagram with the positions of the studied systems. The parameters of the components of the 2 Lac system were calculated using the mass ratio derived from the TCO orbital solution. Evolutionary tracks for single rotating stars are taken from [28] and shown by solid lines along with the location of the zero-age main-sequence. Numbers by the tracks indicate initial masses in solar units.

### 6.2.1. Mizarb

Based on the fundamental parameters of the primary component of MizarB (Table 4), its evolutionary mass is 1.8 M$_\odot$ (Figure 5). From the mass function calculated for MizarB and the absence of signs of the secondary component, one can conclude that the secondary is a very low-mass star. Its parameters depend on the unknown orbital inclination angle $i$. It would be a K3 V star with a mass of $0.77 \pm 0.03$ M$_\odot$ at $i = 20°$ or an M3 V star with a mass of $0.37 \pm 0.02$ M$_\odot$ at $i = 40°$. In both cases, the secondary would be much fainter than the primary component (at least 5 mag visually and at least 2.4 mag in the $K$ band). We were unable to find IR photometry of MizarB because of its close projectional distance to MizarA that results in the IR brightness reported for both components together.

Due to the orbital eccentricity, the maximum angular distance between the MizarB system components at apastron comes to 16–32 milliarcseconds for the inclination angles of 20°–40°, respectively. Although such angular distances are within the reach of modern days interferometry, it would be hard to resolve the components due to the large brightness ratio, which increases at larger inclination angles.

### 6.2.2. $\phi$ Aql

The situation with $\phi$ Aql is similar (no signs of the secondary component in the spectrum). Furthermore, neither optical nor IR photometry (between 0.36 and 22 μm) shows any deviation from the spectral energy distribution expected for an A1 V star. Based on the fundamental parameters of the primary component, its mass is 2.2 $M_\odot$ (Figure 5). It follows from the mass function that the secondary component would be a K1 V to M1 V star for the inclination angles between $40°$ and $90°$, respectively.

Photometric data taken by the TESS mission between 8 July 2022 and 4 August 2022 with a 2-min cadence show brightness variations with an amplitude of 2.2 mmag modulated with the orbital period. Minima occur at phases 0.0 and 0.5 and maxima occur at phases 0.25 and 0.75 (see Figure 3). Such a behavior may be due to grazing eclipses. Considering average masses and radii of K and M dwarfs (e.g., [29]) along with those of the primary component, one can calculate that a critical inclination angle at which eclipses become observable is $79°$. This constrains the mass of the secondary at 0.5 $M_\odot$ that corresponds to an early M–type dwarf.

The VAST survey of all A–stars within 75 pc [30] detected a third component of the $\phi$ Aql system—a faint star at a separation of 2.83 arcseconds (190 AU, projected) with an orbital period of more than 1500 years. The VAST study also reports the detection of X–ray emission in the $\phi$ Aql system, which may be originated from the late-type secondary component. Short-term brightness variations near the orbital phases 0.2 and 0.5 may be due the secondary's surface activity.

### 6.2.3. 2 Lac

Only fundamental parameters of the primary component of the 2 Lac system have been previously published except for the mass ratio. The latter was found to be very similar in all the papers discussed above as well as in our data ($M_2/M_1 = 0.81 \pm 0.01$). The contribution of the secondary component to the system brightness has not been taken into account in the published calculations of the fundamental parameters. Only Petrie [31] derived a visual brightness difference of $\Delta m = 1.03$ mag from some spectral line profiles and listed the components' spectral types as B5 and B6.

Using our spectra at orbital phases where it was possible to measure some of the components' spectral lines separately, we calculated the equivalent width ratios for 9 narrow lines (Fe II 4233 and 5169 Å, C II 4267 Å, He I 4713, 5876 and 6678 Å, Mg II 4482 Å, and Si II 6347 and 6371 Å). We also used the equivalent width ratios of the helium and silicon lines to estimate the components' $T_{eff}$ (see [32] for description). Although both components contribute to the total brightness of the system, their effective temperatures are close to one another and the equivalent width ratios can be used as the first approximation to estimate their fundamental parameters. Our data show that the primary component has $T_{eff} \sim 15,000$ K in agreement with that derived in [33], where it is listed as 15,250 K from the bolometric flux method and 14,900 K from the Balmer jump method.

According to our spectroscopic data, the secondary component has slightly stronger He I 4713 and 5876 Å lines with respect to the Si II 6347 Å line. Furthermore, the equivalent width ratios of the primary and secondary component lines were found to be systematically larger as the wavelength increases (from $\sim$2.5 at 4233 Å to $\sim$3.5 at 6678 Å). Both these types of measurement show that the secondary component of the 2 Lac system is slightly hotter than the primary component. We estimate its $T_{eff} \sim 16,000$ K. Therefore, the secondary component can be classified as a B5 V star, while the primary component is a B6 V star.

Using the above mentioned components' $T_{eff}$, corresponding bolometric corrections [29], the Gaia DR3 distance (see Table 4), and applying various fractional contributions to the observed $V$–band brightness to derive their individual luminosities, we searched for the best match of the evolutionary mass ratio and the spectroscopic one. The best-fit contributions turned out to be 80% for the primary and 20% for the secondary component. This result leads to the components' individual brightness of $V_1 = 4.78$ mag and $V_2 = 6.29$ mag, for the brighter and fainter component, respectively. The secondary component is 1.5 mag fainter

than the primary that is somewhat larger than that found by Petrie [31]. The components' luminosities are then $\log L_1/L_\odot = 3.06 \pm 0.03$ and $\log L_2/L_\odot = 2.53 \pm 0.03$ (the errors are due the distance uncertainty) and the evolutionary masses are 5.1 $M_\odot$ and 4.2 $M_\odot$, respectively, (see Figure 5).

In order to check the components' fundamental parameters we calculated model atmospheres for a range of effective temperatures ($T_{eff}$ = 12,000–17,000 K), surface gravities ($\log g$ = 3.5–4.5), and rotational velocities ($v \sin i$ = 20–50 km s$^{-1}$) using the code *SPECTRUM* [34] that allows calculation of absorption-line spectra, combined them with model spectral energy distributions in the continuum (normalized to the flux at $\lambda = 5500$ Å, [35]), shifted the components' model spectra according to the RV measured at phases of a large separation, and combined them together with various relative contributions. The solar chemical composition was used for the modeling of both system components. A comparison between such a model spectrum and an observed spectrum is shown in the right panel of Figure 1 with the model atmosphere parameters of both components described in Figure caption.

The inclination angle of the system which can be calculated from the components' mass functions or $m \sin^3 i$ values (see Table 3) and the evolutionary masses turns out to be $i = 30 \pm 5°$. A very similar result follows the assumption about the pair synchronization. The averaged orbital velocities of the components are 97 km s$^{-1}$ for the primary and 46 km s$^{-1}$ for the secondary. These combined with the inclination angle imply the projected rotational velocities of $v \sin i$ = 49 km s$^{-1}$ for the primary and 23 km s$^{-1}$ for the secondary, which very closely reproduce the spectral line widths (see right panel of Figure 1).

TESS observed 2 Lac from 11 September 11 to 4 October 2019 and from 22 June 2022 to 21 July 2022. The light curves show orbital modulation with an amplitude of $\sim$2 mmag and are not the same from cycle to cycle. The maxima occur at phases 0.25 and 0.75, while minima occur at phases 0.0 and 0.5. This behavior is consistent with effects of the surface temperature gradient from pole to equator of both components due to their gravitationally distorted shapes.

## 7. Conclusions

CCD spectroscopic observations of three binary systems (MizarB, $\phi$ Aql, and 2 Lac) with an échelle spectrograph at a spectral resolution of $R \sim$12,000 taken at TCO mostly in 2018–2022 allowed to improve the orbital solutions in comparison with those derived from photographic spectra taken in 1908–1972. The modern data have larger signal-to-noise ratios and a better wavelength calibration that result in less scatter of the data points around the best fitting theoretical RV curves folded with the orbital periods. We also used a modern approach to the analysis of the orbital parameter errors and found that the previously published errors were underestimated in most cases.

The use of recent data for the systems' distances, photoelectric photometry, and effective temperatures, allowed us to suggest the range of the secondary component parameters for MizarB and $\phi$ Aql and improve estimates for the secondary component of 2 Lac. In particular, it turned out that the latter is slightly hotter than the primary component. Higher-resolution spectroscopy should allow to separate more spectral lines in the spectrum of 2 Lac and that would lead to tighter constraints on the fundamental parameters of both system components.

This experience calls for high-resolution spectroscopic observations of more bright binary systems, which have been abandoned several decades ago due to improved sensitivity of the modern light detectors and only observed with photographic plates. Our results suggest the value of high-resolution spectroscopic observations with modern techniques of other bright binary systems. Many of these stellar systems have not been studied for several decades. Substantial improvements in technique and the increased sensitivity of modern light detectors over photographic plates will pay substantial dividends. Such a project does not require large apertures and can be conducted with small professional and even amateur telescopes.

**Author Contributions:** Observations, A.S.M., S.D. and A.N.A.; Data reduction, A.S.M.; Data analysis, A.S.M., S.D., H.B. and D.L.; Software I.L.A. and A.S.M.; writing—original draft preparation A.S.M., I.L.A. and H.B.; writing—review and editing A.S.M., S.D., I.L.A. and A.N.A. All authors have read and agreed to the published version of the manuscript.

**Funding:** This research received no external funding.

**Data Availability Statement:** The original spectra reported in this study are available on request to the first author via email at a_mirosh@uncg.edu.

**Acknowledgments:** We acknowledge technical support Dan Gray (Sidereal Technology company), Joshua Haislip (University of North Carolina Chapel Hill), and Mike Shelton (University of North Carolina Greensboro) as well as the Three College Observatory funding by the College of Arts and Sciences and Department of Physics and Astronomy of the University of North Carolina Greensboro. This paper includes data collected by the TESS mission, which are publicly available from the Mikulski Archive for Space Telescopes (MAST). Funding for the TESS mission is provided by NASA's Science Mission directorate.

**Conflicts of Interest:** The authors declare no conflict of interest.

## Abbreviations

The following abbreviations are used in this manuscript: RV—radial velocity, R—spectral resolving power, TCO—Three College Observatory, CCD—charge-coupled device, DAO—Dominion Astrophysical Observatory.

## Note

1. https://www.shelyak.com (accessed on 20 December 2022)

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
