# Peer review of "Refining Orbits of Bright Binary Systems"

_galaxies, doi:10.3390/galaxies11010008_

Round 1

Reviewer 1 Report

The paper under review is devoted to a very interesting and promising observational type of binary systems, the spectroscopic binaries.
From the observation of such systems it is possible to obtain the parameters of orbits and components.
The authors have studied three bright systems of this type and obtained new, improved parameters for them.
The results obtained by the authors will be in demand by the researchers in binary stars astronomy.
I believe that the paper can be accepted for publication by the Galaxies journal after several circumstances have been clarified.

> Dealing with single-lined binaries is more complicated, because the inclination angle of the system's rotational axis needs to be known to constrain the masses of both components. In double-lined binaries this parameter can be found from the components' mass ratio, which is determined by the inverse ratio of the RV amplitudes (Introduction, lines 24-27).

This is an incorrect statement.
To determine the masses of both components (m1, m2), the inclination angle of the system's rotational axis (i) needs to be known _for double-lined binaries as well_.
What one can calculate for double-lined binaries is m1*sin(i)^3 and m2*sin(i)^3.
The components' mass ratio (m1/m2, determined from the ratio of the RV amplitudes) does not help us to obtain individual masses.
As to _single-lined binaries_, here even the knowledge of the inclination angle does not allow one to calculate individual masses.
The only parameter, which can be calculated is so called mass function, f(m) = (m2*sin(i))^3 / (m1+m2)^2.

As I can see, these are the values of f(m) presented in Table 1 and 2.
However, I do not understand which values are presented as f(m) in Table 3 (for 2 Lac).
This is a double-lined binary, the authors measured both RV semi-amplitudes K1 and K2, hence the values m1*sin(i)^3 and m2*sin(i)^3 may well be represented, which is more valuable than just f(m).

Reviewer 2 Report

It is interesting to know there are still room for improvement of the orbit determination of those brightest binaries. It is a laboursome work to observe those stars with high resolution spectra especially when the period is long. The orbit solution show some improvements with the new data, yet there is no detail description about method of the estimation of fundamental parameters, which make the conclusion less reliable. I suggest the author to reconsider section 6.2, show more details and combine their spectral data with the latest survey results, that may make the fundamental parameter estimation more reliable and significant.

The major issues:

1 For Mizar B and Phi Aql, no description of the mass of the primary could be found in the paper, which make the mass or spectral type of the secondary less reliable. Statement like page 7 line 176-177, the author didn’t show how the mass and the error of the secondary are derived without given the primary mass , error and the corresponding method.

2. For 2 Lac , the temperature of the 2 components are derived by measuring the EW and the ratio of EWs,  which is questionable since the measurement of EW is based on the relative width of the continuum. When the target is SB2 both star contribute to the continuum, the measured EW will be smaller than the true value, which makes the subsequence deduced temperature, flux ratio less reliable. The author should show detail of their EW estimation especially how they deal with the continuum and relative contribution of the 2 stars and how the subsequence parameter are determined. An alternative way to determine the parameter of the components is to use the spectral disentangle technic like FDbinary(Ilijic 2004).

Some minor issues:

1.     Page1. introduction, line 25-26,”In double-lined binary this parameter can be found”, It is not clear what “this parameter” means, if “this parameter” refer to inclination, I don’t see how the inclination could be derived with RV curve, please make more clear explanation. The same detail description should be given at page 8 line 226.

2.     Page 2, section 3, observation and data reduction. The exposure time and typical S/N of each observation should be listed to let the reader evaluate the corresponding RV error in the figure.

3.     Page 2, Section 4, Results.  There should be a figure demonstrate the spectrum of each star, the corresponding lines mentioned in the paper should be marked.

4.     Page3, line 98 HR1389 is a K giant, which is different from A type star, please evaluated the RV error induced by using model of different spectral type.

5.     Page 4, Table 2 footnote, “…. Except for TO,“ TO should be T0.

6.     Page 5, equation 1,  please check the superscript of the Summation, It is a layout problem.

7.     Page 7, table 4,  It is strange why the author won’t use the basic stellar parameters such as Teff Logg and [Fe/H] based their own data, instead of those with those from literature. A comparison could be made.

8.      Page 8, line 208, those lines should be mark in a figure for a direct perception.

Suggestions:

1. There are new photometric light curves for TESS for phi Aql and 2 Laccombining the RV curve with light curve may reveal more physical character of those systems.

2. For 2 Lac, assuming the 2 component are synchronized, the inclination could also be derived by measuring the spin velocity, vsini , which should be resolved with resolution 12000.

Round 2

Reviewer 2 Report

I don't think the question of 2 Lac was solved properly.  The line strength between the data and the model read from the right panel of figure 1 are not consistent. The model and the data should be put together for comparison or a residual should be shown at the bottom.The  author should show  at least a figure of the Chi-square distribution between the data and different models or how they converge at 80% and 20% flux ratios and 16K vs 15K with whatever method.  

I don't understand why the author says the 2 components of 2 Lac are not synchronized,  please explain.  

Round 3

Reviewer 2 Report

no further suggestions or comments